# A transcriptome based molecular classification scheme for cholangiocarcinoma and subtype-derived prognostic biomarker

Zhongqi Fan[1,6], Xinchen Zou[2,6], Guangyi Wang[1], Yahui Liu[1], Yanfang Jiang[3], Haoyan Wang[2], Ping Zhang[1], Feng Wei[1], Xiaohong Du[1], Meng Wang[1], Xiaodong Sun[1], Bai Ji[1], Xintong Hu[3], Liguo Chen[3], Peiwen Zhou[3], Duo Wang[3], Jing Bai[2], Xiao Xiao[4], Lijiao Zuo[2], Xuefeng Xia[2], Xin Yi[2,5] & Guoyue Lv[1] ✉

Previous studies on the molecular classification of cholangiocarcinoma (CCA) focused on certain anatomical sites, and disregarded tissue contamination biases in transcriptomic profiles. We aim to provide universal molecular classification scheme and prognostic biomarker of CCAs across anatomical locations. Comprehensive bioinformatics analysis is performed on transcriptomic data from 438 CCA cases across various anatomical locations. After excluding CCA tumors showing normal tissue expression patterns, we identify two universal molecular subtypes across anatomical subtypes, explore the molecular, clinical, and microenvironmental features of each class. Subsequently, a 30-gene classifier and a biomarker (called "CORE-37") are developed to predict the molecular subtype of CCA and prognosis, respectively. Two subtypes display distinct molecular characteristics and survival outcomes. Key findings are validated in external cohorts regardless of the stage and anatomical location. Our study provides a CCA classification scheme that complements the conventional anatomy-based classification and presents a promising prognostic biomarker for clinical application.

Cholangiocarcinoma (CCA), which is the second most common liver cancer and constitutes ~15% of all primary liver malignancies[1], has currently been a significant public health concern worldwide. Unfortunately, the global incidence and mortality of CCA have continued to increase in recent years[2,3]. CCAs can arise anywhere in the biliary tree and are typically classified into intrahepatic (iCCA), perihilar (pCCA), and distal (dCCA) subtypes based on their anatomical origin[4].

Despite the widespread use of this conventional anatomical-based classification, it has several limitations. For example, CCA (especially pCCA) is often diagnosed at an advanced stage, making it challenging to distinguish between intrahepatic or extrahepatic locations[5]. Additionally, there are no apparent pathological or molecular differences between pCCA and the iCCA with large intrahepatic bile ducts as the source[6]. While various molecular targeted treatments have been approved (e.g., FGFR-1/2/3 inhibitors[7,8] for treating iCCA), surgical resection remains the primary choice for early-stage CCA, and treatment options are limited[9]. Therefore, it is critical to conduct comprehensive molecular studies of CCA to gain a deeper understanding

[1]Department of Hepatobiliary and Pancreatic Surgery, General Surgery Center, The First Hospital of Jilin University, Changchun, China. [2]Geneplus-Beijing Institute, 9th Floor, No.6 Building, Peking University Medical Industrial Park, Zhongguancun Life Science Park, Beijing, China. [3]Genetic Diagnosis Center, The First Hospital of Jilin University, Changchun, China. [4]Geneplus-Shenzhen, No.14 Zhongxing Road, Pingshan District, Shenzhen, China. [5]School of Computer Science and Technology, Xi'an Jiaotong University, Xi'an, China. [6]These authors contributed equally: Zhongqi Fan, Xinchen Zou. ✉e-mail: lvgy@jlu.edu.cn

of its molecular mechanisms and pathogenesis. Such studies could aid in developing classification schemes and targeted therapies.

Nevertheless, previous studies on the molecular classification of CCA have been limited to a certain anatomical site[6,9–11]. For example, Montal et al. performed an integrative molecular analysis of 189 extrahepatic CCA tumors and identified four molecular subtypes[9], while Sia et al. identified two molecular classes of iCCA with different survival outcomes[10]. Though these studies provide valuable therapeutic choices, the developed classification schemes rely largely on accurate annotation of the anatomical site of CCA, and some have not accounted for normal tissue contamination biases in transcriptomic profiles. However, in the clinic, it was difficult to require entirely pure anatomical areas and most were mixed with other site contamination. This could skew the classification outcomes and affect treatment options.

To address these limitations, we performed whole-genome expression profiling on 438 CCA patients. By analyzing tumors that avoided obvious tissue contamination, we identified two universal molecular subtypes of CCA across anatomical locations that showed distinct overall survival. We report on the clinical and molecular features of these classes and the development of a prognostic scoring system based on CholangiOcarcinoma anatomy-independent RNA Expression index including 37 genes (CORE-37).

## Results

### Clinicopathological characteristics of CCA patients

This study enrolled 438 patients diagnosed with either dCCA, pCCA, or iCCA (Table S1). The majority of patients underwent surgical resection, with 5 and 3 patients receiving surgical biopsy and puncture biopsy, respectively. The median age of the participants was 63 (range, 25–82), and 67.35% of them were male. With regards to anatomical location, 43.15% were diagnosed with dCCA, 30.82% with pCCA, and 26.03% with iCCA. Pathological TNM stage differed by anatomical locations (Table S1), with IIA/IIB in dCCA, II/IIIC in pCCA, and II/IIIB in iCCA.

### Initial transcriptome-based CCA subtypes are biased with tissue contamination and anatomical location

Based on the technical criteria of pathological specimen collection, in order to preserve the spatial relationship information between tumor tissues and adjacent organs to facilitate pathologists in determining the origin of CCAs and infiltration degree, some FFPE samples unavoidably contain mixed normal tissues, especially in dCCA (Fig. S1). To evaluate how tissue contamination and anatomical location may impact molecular subtype classification, we initially conducted an unsupervised clustering analysis on the transcriptomic data using NMF method ($N = 438$; Table S1, Table S2, Fig. S2). Our analysis revealed four molecular subtypes named C1-4 (Fig. S2, Fig. 1a). We observed significantly enriched iCCAs in C1 and C4, pCCAs in C3, and dCCAs in C2 (Fig. 1a). Normal tissue contamination significantly impacted molecular classification, with hepatic tissue more predominant in C4 and pancreatic tissue in C2 (Fig. 1a, Table S3). The subtype preference was also reflected in the distribution of duodenal, lymphatic and neural tissues (Fig. 1a, Table S3). Since the anatomical location of the lesions determines the surgical approach and the extent of tissue contamination, we hypothesized that the bias in this molecular typing is mainly due to the proportion and type of contaminated tissues in CCAs. Overall, our initial transcriptome-based classification was biased by the effect of tissue contamination.

### Transcriptomic profiling of selected samples identifies two universal CCA subtypes across different anatomical locations

To reduce the impact of tissue contamination on molecular classification of CCAs and find a molecular classification scheme that independent on anatomical locations, we selected samples with low overall contamination proportions from 438 samples for model construction,

while the remaining samples were used as verification cohort. The overall contamination ratio for each sample was estimated by two independent pathologists (overall contamination ratio = non-cancer tissue area /total tissue area ratio, including hepatic, pancreatic, duodenal, lymphatic and neural tissues contamination). Given that hepatic and pancreatic contamination were the most predominant in our cohort, we collected previously published liver-specific and pancreas-specific gene markers as templates for NTP analysis (Table S4)[12]. After filtrating out samples based on NTP results and overall contamination proportions (see "Material and Methods"), we retained a total of 164 samples which is referred to as the "purified cohort" for the subsequent analyses, while the other 274 samples were used as the "verification cohort" (Fig. 1b, Table S1).

Hierarchical clustering analysis on the purified cohort showed that CCAs with low contamination degree failed to be distinguished by anatomical location at the transcriptomic level (Fig. 1c), likely due to low anatomy-related specificity on their transcriptomic profiles. We thus aimed to explore anatomy-independent molecular classification scheme, and employed a consensus clustering approach on the purified cohort. The results strongly supported the existence of two clusters (Fig. 1d, Fig. S3). Based on the expressed molecular characteristics (detailed later in the text), we categorized these two subtypes as "Mesenchymal & Immunosupressive-C1" (35.37%) and "Metabolic & Proliferative-C2" (64.63%). Importantly, we found no significant correlation between the identified molecular subtype and the prevalence of tissue contamination as observed in our initial results (Fig. 1d, Table S5), except for hepatic contamination, which was significantly more prevalent in the Metabolic & Proliferative-C2 subtype than in Mesenchymal & Immunosupressive-C1 subtype (Table S5). Additionally, our analysis of clinical-anatomical characteristics showed no association between molecular subtype and anatomical location (Table S5), supporting our hypothesis that the previously observed significant correlation between molecular subtypes and anatomical locations was affected by the uneven distribution of tissue contamination across anatomical sites. In summary, we identified two universal subtypes across anatomical locations by focusing on the purified cohort of 164 selected CCA tumors.

### Transcriptional analysis reveals distinct molecular features and tumor immune evasion mechanisms between subtypes

We performed gene set enrichment and expression deconvolution analyses to comprehensively assess the molecular features and tumor microenvironment of each subtype.

Mesenchymal & Immunosupressive-C1: accounts for 35.37% of the purified cohort (58/164) and exhibited high activity of the epithelial-mesenchymal transition (EMT) pathway (Fig. 2a). The aberrant activation of EMT-related, hedgehog and TNFα signaling pathways, which previously reported in the mesenchymal eCCA class[9], was observed in this subtype (Fig. 2a). This was in line with a relatively higher stromal score (Fig. 2b). In addition, immunohistochemistry (IHC) was performed on tissue microarrays to verify the expression level of key epithelial-mesenchymal transition proteins. E-cadherin is significantly lower in C1 than in C2, while Vimentin showed higher expression in C1 tumor cells compared to C2, indicating a stronger epithelial-mesenchymal transition in C1 than C2, although N-cadherin did not show significant differences between the two groups (Fig. S4). Additionally, this subtype showed high signals of angiogenesis known to promote tumor growth and invasiveness, with high activity of the hypoxia pathway and significantly higher hypoxia signature score (Fig. 2a, c, Fig. S5). It has been reported that hypoxia can induce the hypoxia-inducible factors (HIF) system to mediate angiogenesis process[13]. Consistently, the subtype had higher mRNA levels of genes involved in HIF system, such as *VEGFA*, *VEGFC* (Vascular Endothelial Growth Factor A and C), and *HIF1A* (Hypoxia-Inducible Factor 1 Subunit Alpha) (Fig. 2d–f, Fig. S1).

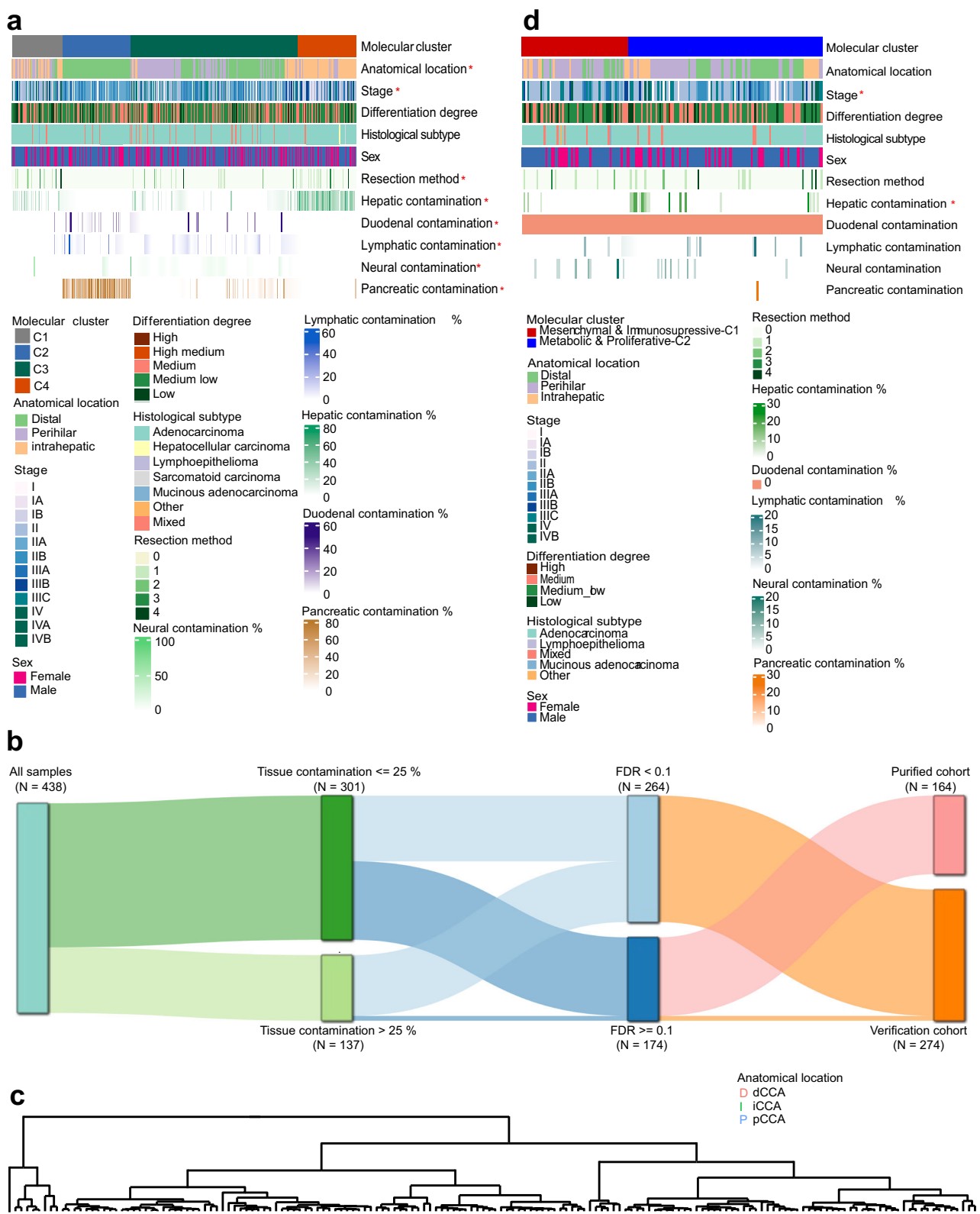

This subtype was partially categorized as "immunosuppressive" due to the presence of enriched classical oncogenic signaling pathways that are linked to immune evasion, including IL-6/JAK/STAT3 and Wnt/β-catenin (Fig. 2a)[14,15]. Several related genes, including *IL-6*, *JAK1*, *JAK2* and *STAT3*, were overexpressed in this subtype (Fig. S6A–D). Activation of the Wnt/β-catenin pathway is known to create non-inflammatory tumor microenvironment by (1), enhancing the survival of regulatory T (Treg) cells; and (2), interacting with tumor-associated

macrophages (TAMs) through Snail (a Zinc finger protein encoded by Wnt-regulated gene), which can in turn increase β-catenin activity via IL-1β[15–17]. Consistent with this, we observed overexpressed *SNAI1* and *IL1B* genes and predominant infiltration of TAMs and Treg cells (Fig. S6E, F, Fig. 2g, h). CD163 was used as marker for M2 macrophages, and it was shown that CD163 expression in C1 was significantly higher than that in C2. On the other hand, CD8 T cells were found to be more abundant in C2; However, this difference did not reach statistical

**Fig. 1 | Correlation between CCA molecular subtypes and clinical and anatomical features. a** Initial non-negative matrix factorization (NMF) clustering on transcriptomic data of all samples (*N* = 438) revealed four molecular classes. Molecular cluster & Anatomical site, *p* = 1.00E-06; Molecular cluster & Stage, *p* = 1.00E-07; Molecular cluster & Resection method, *p* = 0.0001; Molecular cluster & Hepatic contamination, *p* = 2.20E-16; Molecular cluster & Pancreatic contamination, *p* = 2.20E-16; Molecular cluster & Duodenal contamination, *p* = 4.20E-06; Molecular cluster & Lymphatic contamination, *p* = 0.0036; Molecular cluster & Neural contamination, *p* = 0.0017. **b** Sankey diagram representing the select rules of the "purified cohort" (*N* = 164) and "Verification cohort" (*N* = 274). First panel: all 438 CCA cases enrolled in this study; second panel: samples grouped according to whether the overall tissue contamination proportion >25%; third panel: samples grouped according to whether the NTP result < = 0.1; fourth column: samples with FDR < = 0.1 in NTP result and with an overall tissue contamination proportion >25% were selected into the "verification cohort", while the other samples selected into the "purified cohort". **c** Dendrogram generated based on expression matrix of the purified cohort (protein-coding genes only; *N* = 164). **d** Two molecular classes were determined through consensus clustering on the purified cohort (*N* = 164). Molecular cluster & Stage, *p* = 0.03; Molecular cluster & Liver percentage, *p* = 0.02; *P* values were calculated by two-sided Fisher's exact test for categorical variables and Kruskal–Wallis rank sum test for categorical and continuous data. Red asterisks indicate variables significantly correlated with molecular classes (*p* < 0.05).

significance (*p* = 0.08), possibly due to small sample size (Fig. S5). Moreover, we observed significantly higher T cell exclusion score in this subtype, consistent with low infiltration level of CD8[+] T lymphocytes (Fig. 2i, j, Fig. S5). Overexpression of genes encoding co-stimulators/co-inhibitors occurred (Fig. 3a), implying possible immune evasion mechanisms due to overexpression of immune checkpoints[17]. These immune-related mechanisms together could be a particular immune evasion strategy employed by this CCA subtype. Treatment with immune checkpoint inhibitors such as CTLA-4 and PDL1 (encoded by *CD274* gene) blockade may be effective for this subtype. Radiotherapies and chemotherapies may also be of benefit to treating this CCA subtype as they have been reported to improve physical properties of tumor microenvironment and improve the recruitment of T cells in tumor tissues[18–20].

Metabolic & Proliferative-C2: is more prevalent in the purified cohort (64.63%) and was categorized by enriched metabolism-related hallmarks such as fatty acid, bile acid and xenobiotic metabolism (Fig. 2a). Genes encoding the peroxisome, an oxidative organelle involves in lipid metabolism, were found overexpressed (Fig. 2a)[21]. ADH1A and CYP3A4 were used as markers to detect metabolic changes by immunohistochemistry (IHC). Consistent with the transcriptome analysis results, we found that both ADH1A and CYP3A4 were significantly lower in C1 compared to C2 (Fig. S5). These features align with those showed in previously reported eCCA Metabolic class[9]. The Metabolic & Proliferative-C2 subtype also exhibited high mRNA levels of genes encoding MYC and E2F targets and activation of the G2M checkpoint pathway, suggesting enriched proliferation-related features (Fig. S5). We observed a predominant enrichment of PI3K-AKT-mTOR and mTORC1 oncogenic pathways, which have been reported associated with ferroptosis resistance in cancer cells[22]. Ferroptosis is a mechanism remaining largely unknown in CCA, which inspired us to explore if this process varied among molecular subtypes. As expected, the ferroptosis-related genes and ferroptosis signature score were dramatically lower in this subtype (Fig. 3b, c), demonstrating its ferroptosis-resistant feature. Interestingly, our molecular subtypes exhibit great similarity with two ferroptosis-related subtypes that were previously discovered in hepatocellular carcinoma[23], encompassing several characteristics, such as pathway enrichment, tumor immune microenvironment, and survival outcomes specific to each subtype (discussed later).

This subtype exhibited relatively high infiltration level of CD8[+] T lymphocytes, CD4[+] T memory resting cells and B naïve cells (Fig. 2j, Fig. 3d, e), though the overall lymphocytic infiltration inferred from transcriptional profiles did not differ significantly between the two subtypes (Fig. 3f). Importantly, we found dramatically low expression levels of MHC-I and MHC-II genes in these CCAs (Fig. 3a), implying a defective antigen presenting and processing capacity of tumor cells as a possible intrinsic immune evasion mechanism of this subtype. Chemotherapies and radiotherapies may also be effective in treating this subtype of CCAs as they can promote antigen-presenting cell recruitment and boost antigen or receptor expression on the membrane of tumors or immune cells[20].

## Survival analysis shows strong differences in survival between two molecular subtypes

We examined the prognostic relevance of our molecular subtypes by employing patients from our purified cohort who had available survival information. After excluding patients with other cause of death, we found that patients in the Mesenchymal & Immunosupressive-C1 class had significantly poorer survival outcomes (Fig. 4a; log-rank sum test, *p* = 1.9 × 10[-4] < 0.05; median OS: 376 vs 565 days in the Metabolic & Proliferative-C2 class), regardless of the anatomical location (Fig. 4b–d). Univariate Cox proportional hazards regression analysis revealed significant association between overall survival and anatomical location, stage and molecular classification (Table 1). Multivariate analysis identified anatomical location (dCCA class as control, *p* = 0.039 for iCCA and 0.166 for pCCA class), stage (*p* = 0.01), and molecular classification (*p* = 0.002) as independent prognostic indicators (Fig. 4e, Table 1). The molecular and clinical features, prognostic relevance, as well as the potential therapeutic strategies in each subtype, are summarized in Fig. 4f.

## Design and validation of CCA classifier confirms the reproducibility of molecular classification

To evaluate the applicability of our classification scheme in independent cohorts, we developed a CCA subtype classifier that based on the impact of genes on signal-to-noise ratio (SNR) scores, genes defining each molecular class were selected to construct the classifier. Our CCA subtype classifier composed of 20 genes defining Mesenchymal & Immunosupressive-C1 and 10 genes defining the other subtype (see "Material and Methods", Fig. 5a, b). Interestingly, 9 out of 10 genes defining Metabolic & Proliferative-C2 were mitochondrial-encoded genes associated with cancer progression (Table S6)[24], highlighting the unknown role of mtDNA overexpression during progression of this CCA subtype. One possible explanation is that activation of cell proliferation and tumor growth in this class demands a large amount of energy and metabolites, leading to enrichment of metabolism-related pathways and overexpression of mtDNA driving mitochondrial respiration and energy production.

This classifier predicted CCA class with a high confidence of 76.8% in our purified cohort (126/164) and an overall precision of 96.03% (121/126) (Fig. 5b). The accuracy of predicting a Mesenchymal & Immunosupressive-C1 sample and Metabolic & Proliferative-C2 sample was 89.66% and 82.08%, respectively. We further performed validation on samples from the verification cohort (*N* = 274: dCCA = 136, pCCA = 51, iCCA = 87), TCGA-CHOL project (*N* = 36: dCCA = 2, pCCA = 4, iCCA = 30)[25], and the Dong study (iCCA = 255)[26]. The validation results from the verification cohort showed that our classifier could confidently classify most CCA tumors with severe tissue contamination into a particular molecular class with overexpression of the corresponding classifier genes (Fig. S7, Fig. S8). In addition, both molecular subtypes were successfully identified in the TCGA-CHOL cohort (Fig. S9, Fig. S10) and the Dong cohort (Fig. S11, Fig. S12, Fig. S13), further revealing the reproducibility of our molecular classification scheme and its potential to be applied to other independent cohorts.

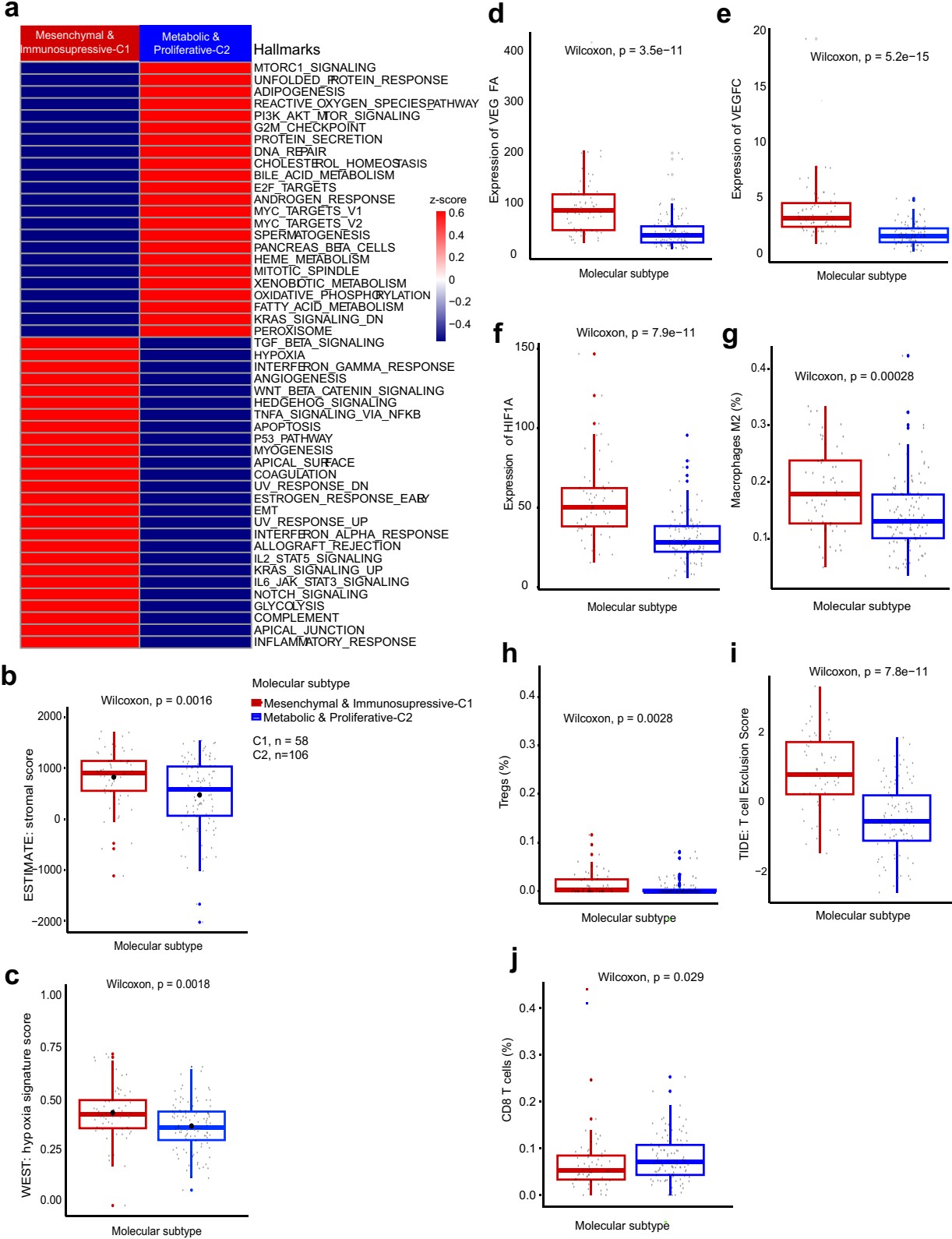

**Fig. 2 | Molecular characteristics and immune microenvironment of CCA classes.** **a** Heatmap representing the enrichment of hallmark gene sets. Single-sample gene set enrichment analysis (ssGSEA) was used to obtain enrichment scores, with samples from the same subtype indicated with a normalized z-score. Box plots representing **b** the estimation of stromal compartment (ESTIMATE package); **c** the estimation of hypoxia signature score (ssGSEA) in each class; **d** relative RNA expression of *VEGFA*; **e** relative RNA expression of *VEGFC*; **f** relative RNA expression of *HIF1A*; **g** the abundance of macrophage M2 (CIBERSORTx); **h** the abundance of regulatory T (Treg) cells (CIBERSORTx); **i** the estimation of T cell exclusion (TIDE software); **j** the abundance of CD8⁺ cytotoxic cells (CIBERSORTx). Box plots show median, interquartile values, range and outliers (individual points), C1, *n* = 58; C2, *n* = 106. *P* values were calculated by two-sided Wilcoxon rank sum test.

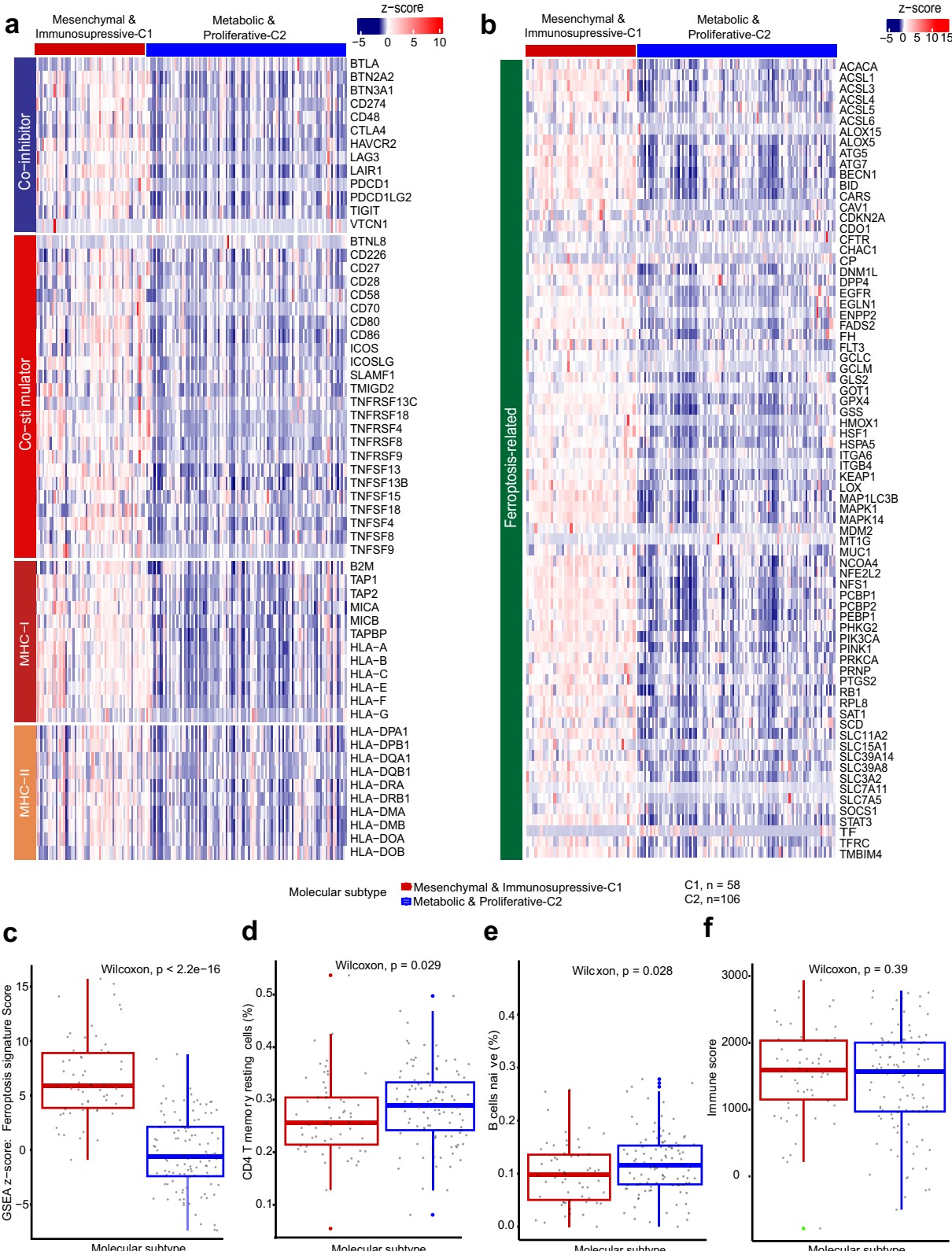

**Fig. 3 | Ferroptosis-resistant features and immune escape mechanisms of CCA classes.** Heatmaps displaying expression levels of **a** genes encoding co-stimulators, co-inhibitors and MHC antigens; **b** ferroptosis-related genes in each molecular class. The expression values were normalized and represented by z-scores. Box plots representing **c** the enrichment of ferroptosis-related gene signature (GSVA applied with z-score method); **d** the abundance of CD4⁺ T memory resting cells; **e** the abundance of B naïve cells; **f** the estimation of immune compartment in each class (ESTIMATE package). Box plots show median, interquartile values, range and outliers (individual points), C1, $n = 58$; C2, $n = 106$. $P$ values were calculated by two-sided Wilcoxon rank sum test.

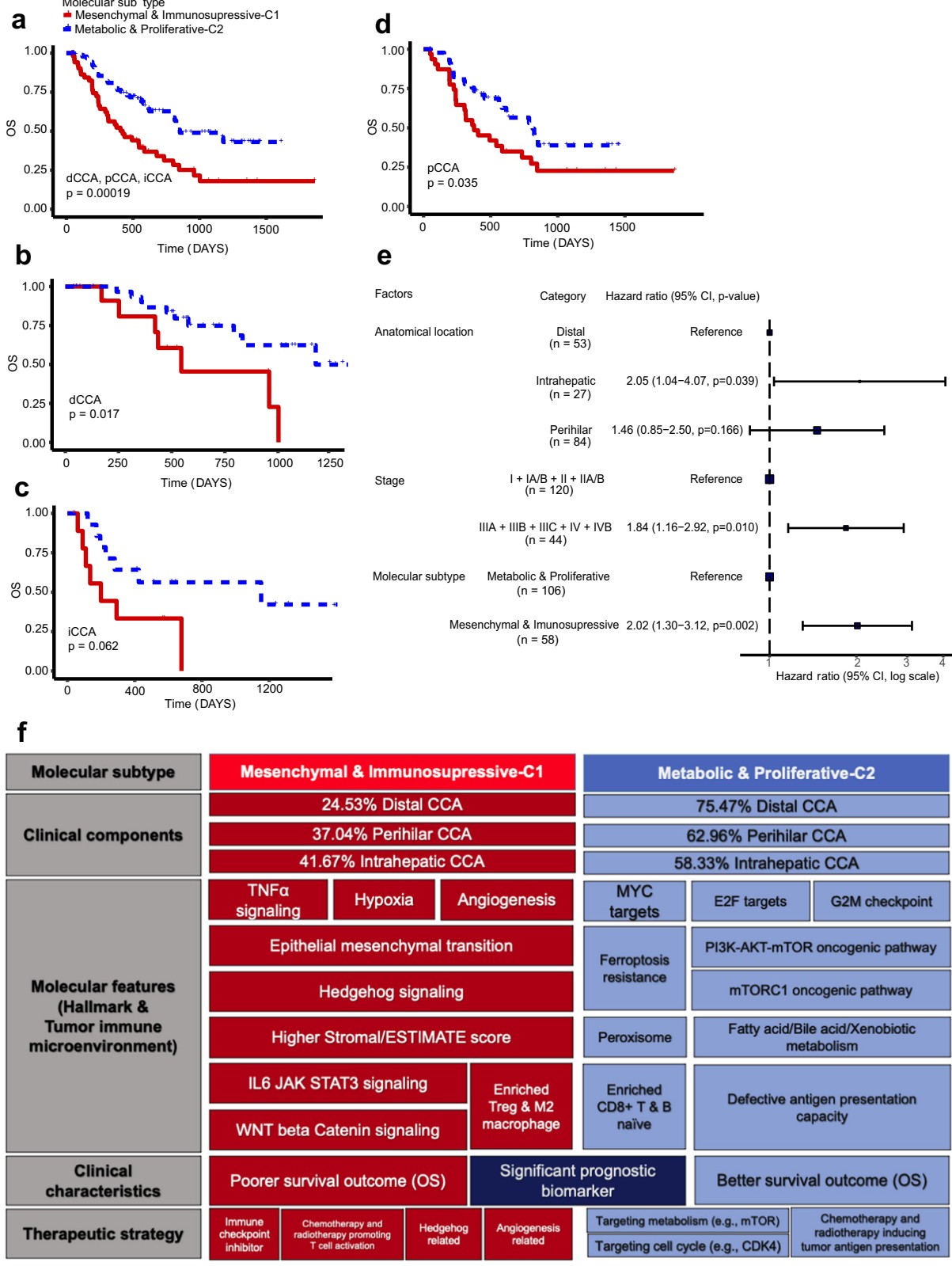

**Fig. 4 | Prognostic relevance and summary of molecular classes.** Kaplan–Meier curves comparing overall survival (OS) between the two CCA classes for **a** all the patients employed in the purified cohort; patients with **b** dCCA; **c** iCCA; **d** pCCA only. *P* values were calculated by log-rank test. **e** Forest plot displaying anatomical location, stage and molecular subtype as independent prognostic indicators. dCCA, *n* = 53, iCCA, *n* = 27, pCCA, *n* = 84; Stage I + IA/B + II + IIA/B, *n* = 120, stage

IIIA + IIIB + IIIC + IV + IVB, n = 44; C1, *n* = 58; C2, *n* = 106. The center of error bars showed observed study effect size. The *p* values are two-sided and *p* < 0.05 was considered significant. **f** Schematic representation of molecular features, prognostic relevance, and potential drug targets in each molecular class. CI confidence interval.

**Table 1 | Univariate and multivariate analysis for overall survival**

| Factor | Category | N (%) | Univariate analysis | | Multivariate analysis | |
|---|---|---|---|---|---|---|
| | | | HR (95% CI) | p | HR (95% CI) | p |
| Age | - | - | 1.02 (0.99–1.05) | 0.176 | | |
| Sex | female | 45 (29.0) | - | | | |
| | Male | 110 (71.0) | 1.54 (0.92–2.56) | 0.1 | | |
| Anatomical location | Distal | **48 (31.0)** | - | | - | |
| | Intrahepatic | **25 (16.1)** | **2.20 (1.12–4.34)** | **0.022** | **2.05 (1.04–4.07)** | **0.039** |
| | Perihilar | **82 (52.9)** | **1.76 (1.04–2.99)** | **0.036** | 1.46 (0.85–2.50) | 0.166 |
| Stage | I + IA/B + II + IIA/B | **113 (72.9)** | - | | - | |
| | IIIA + IIIB + IIIC + IV + IVB | **42 (27.1)** | **2.18 (1.39–3.42)** | **0.001** | **1.84 (1.16–2.92)** | **0.01** |
| Differentiation | Low + Medium low | 98 (63.2) | - | | | |
| | Medium + High | 57 (36.8) | 0.98 (0.63–1.53) | 0.921 | | |
| Molecular subtype | Metabolic & Proliferative-C2 | **99 (63.9)** | - | | - | |
| | Mesenchymal & Immunosupressive-C1 | **56 (36.1)** | **2.20 (1.43–3.38)** | **<0.001** | **2.02 (1.30–3.12)** | **0.002** |

The p values are two-sided and p < 0.05 was considered significant. **Clinical factors** with a p < 0.05 (considered significant) in both univariate and multivariate analysis were highlighted by **bold** style. CI confidence interval, HR hazard ratio.

Next, we investigated if the main molecular features of two subtypes identified from the purified cohort were consistent with those from the verification cohorts and external cohorts. The results from all three validation cohorts showed similar pathway enrichment patterns in each subtype as observed in the purified cohort (Fig. S7B, Fig. S9B, Fig. S11B). The downregulated expression of ferroptosis-related genes and MHC-I/MHC-II genes in Metabolic & Proliferative-C2 and overexpression of genes encoding co-stimulator/inhibitor molecules in Mesenchymal & Immunosupressive-C1 were observed in some cohorts (Fig. S8, Fig. S13). However, divergencies were observed in TCGA-CHOL cohort and Dong cohort (Fig. S10, Fig. S12). These discrepancies may be attributed, in part, to the bias introduced by tissue contamination in both cohorts.

Owing to available proteomic data from the Dong cohort[26], we were able to validate specific expression patterns in each subtype at the proteomic level. As expected, proteins encoded by certain classifier genes were overexpressed in the corresponding subtype (Fig. S11A). The protein levels of MHC-I/MHC-II antigens and co-stimulator/inhibitor molecules were relatively higher in Mesenchymal & Immunosupressive-C1 (Fig. S13), which was consistent with our findings at the transcriptomic level.

Overall, the validations confirmed the utility of our classification scheme, the accuracy of our classifier and the reproducibility of molecular features in each subtype. Given its prognostic relevance and independence on anatomical location and tissue contamination, it may supplement the conventional clinical approach of tumor classification based on anatomical location and aid treatment decision-making.

## A prognostic biomarker derived from differential gene expression analysis between subtypes

As the 30-gene classifier divides samples into two categories, quantifying the relationship between this categorical variable and survival outcomes can be challenging. To address this issue, we sought to identify a continuous variable as a prognostic biomarker derived from subtypes, the value of which is related to survival outcomes in CCA patients. To achieve this, we performed differential expression analysis on our purified cohort to identify differentially expressed genes (DEGs) in each subtype (adj p < 0.01 and absolute fold change>2, Metabolic & Proliferative-C2 class as reference group). We selected 25 DEGs as a C1-like signature and 12 DEGs as a C2-like signature (Table S8). By combing two signatures and employing 'Singscores' method, we were able to quantify C1-like and C2-like status of a tumor and a unified score for inferring its overall status (see "Materials and Methods"). The unified-score-based prognostic evaluation scheme was

termed as "CORE-37" (CholangiOcarcinoma anatomy-independent RNA Expression index including 37 genes). As expected, CCAs from C1 class exhibited predominantly C1-like features and a low C2-like signature score, resulting in a relatively high CORE-37 score, while the opposite situation occurred in CCAs identified as C2 class (Fig. 5c).

We next investigated prognostic relevance of the CORE-37 score. Survival analysis in our purified cohort showed that the CORE-37 score stratified patients in classes with significantly different survival outcomes (Fig. 5D; log-rank sum test, $p = 7 \times 10^{-3} < 0.05$). Multivariant Cox regression analyses further showed that CORE-37 score (either considered as a continuous variable or categorical variable based on quartiles) was an independent prognostic biomarker (Fig. 5e, f). We validated our findings on the verification cohort, Jusakul cohort[27], and Dong cohort[26]. Consistently, all confirmed that the CORE-37 score was an independent prognostic factor (Figs. S14–S16). Moreover, we observed a stepwise increase in hazard ratios with the increase of categorical CORE-37 score in all the cohorts, highlighting its reliability as a prognostic indicator. The dysregulation of the composed genes that compose this score may lead to continuous effects on tumor malignancy.

We further measured the usefulness of CORE-37 biomarker (Fig. 6, Table S9). ROC plots suggested that the prediction performance of CORE-37 biomarker changed as time passed by, with the maximum AUC occurred at year 4 (Fig. 6a). We validated the predictive ability of the CORE-37 biomarker through NRI analysis, we compared the model considering age and TNM stage (as the control/standard model) with the one considering age and CORE-37 score (Fig. 6b, Table S9). As TNM-staging criteria vary between CCAs with different anatomical location, we separately performed comparisons regarding to different anatomical locations. The results showed positive NRI values in all three comparisons, demonstrating that the prediction performance of CORE-37 prognostic biomarker was superior to that of TNM-staging, regardless of anatomical location. Additionally, we estimated the performance of integrated model (involved age, TNM stage, CORE-37 score). The results showed even greater NRI value of this model for dCCAs and iCCAs when comparing with that of model considering age and CORE-37 score (Table S9), implying a better performance of the integrated model for dCCAs and iCCAs.

In summary, the CORE-37 score derived from the molecular subtype signatures is a highly reliable prognostic indicator of CCAs, regardless of stage and anatomical location of tumors. These results validate the prognostic performance and applicability of CORE-37 score. Since our prognostic biomarkers are independent of anatomical locations and tissue contamination, the CORE-37 score can be used to

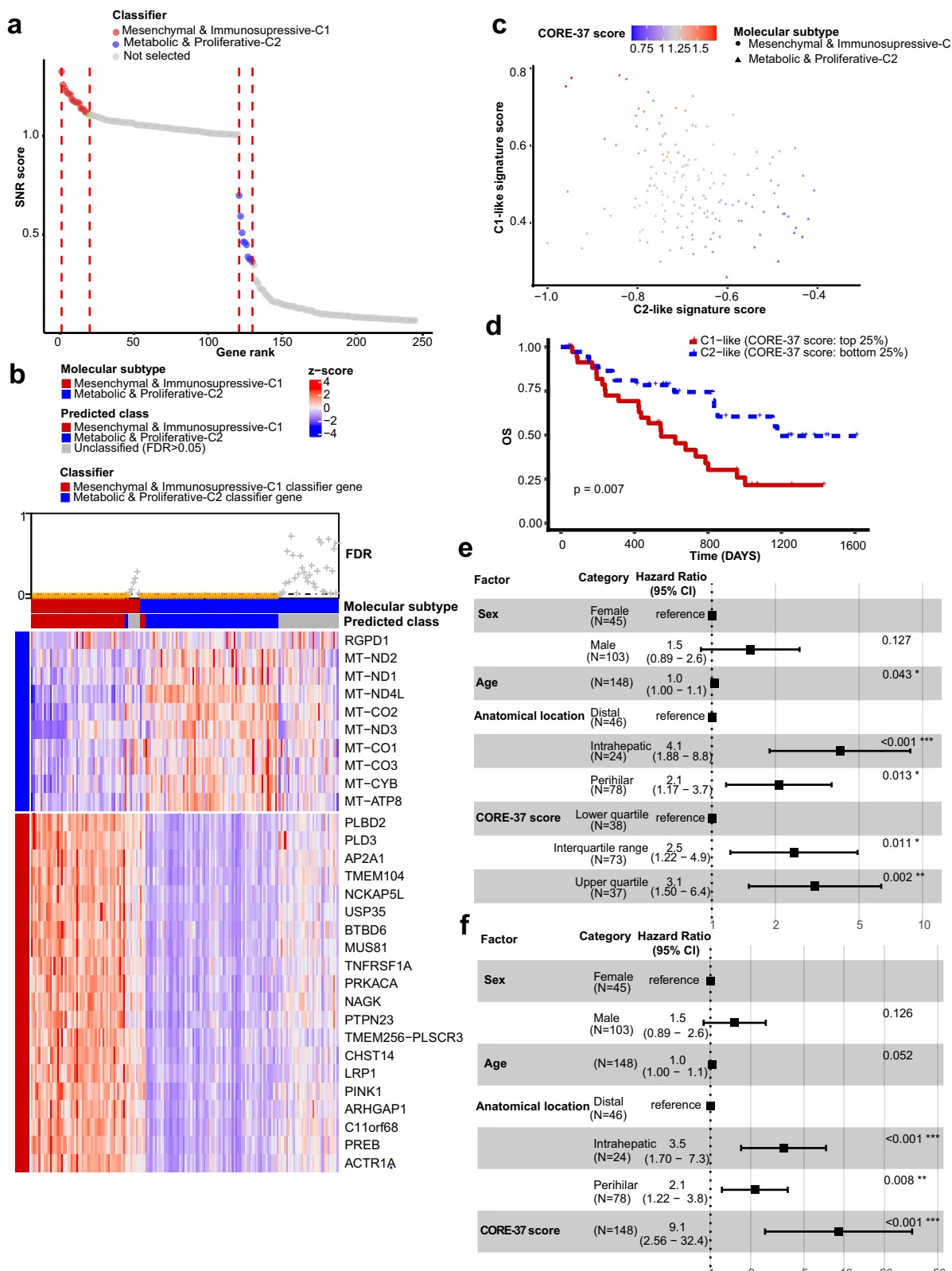

**Fig. 5 | Molecular classifier construction and subtype-derived prognostic indicator development. a** Dot plot representing signal-to-noise ratio (SNR) score versus gene rank (Class Neighbors tool). Genes largely contributing to the decrease of SNR score were selected to construct classifier. **b** Heatmap representing the expression of classifier genes in each sample (purified cohort). The expression levels were represented by normalized z-scores. The predicted molecular class of these samples was obtained by nearest template prediction analysis (NTP), together showed with their original molecular class. **c** Scatter plot displaying C1-like and C2-like signature scores of each sample from the purified cohort, with samples colored by their CORE-37 score and shaped by their molecular subtype. **d** Kaplan–Meier curve comparing overall survival (OS) between samples within the top and bottom quartiles of CORE-37 scores. Forest plots illustrating **e** quartile-categorized CORE-37 score and **f** continuous CORE-37 score as independent prognostic indicators, regardless of age, sex and anatomical location. *P* values were calculated by log-rank test. CI confidence interval. The center of error bars showed observed study effect size.

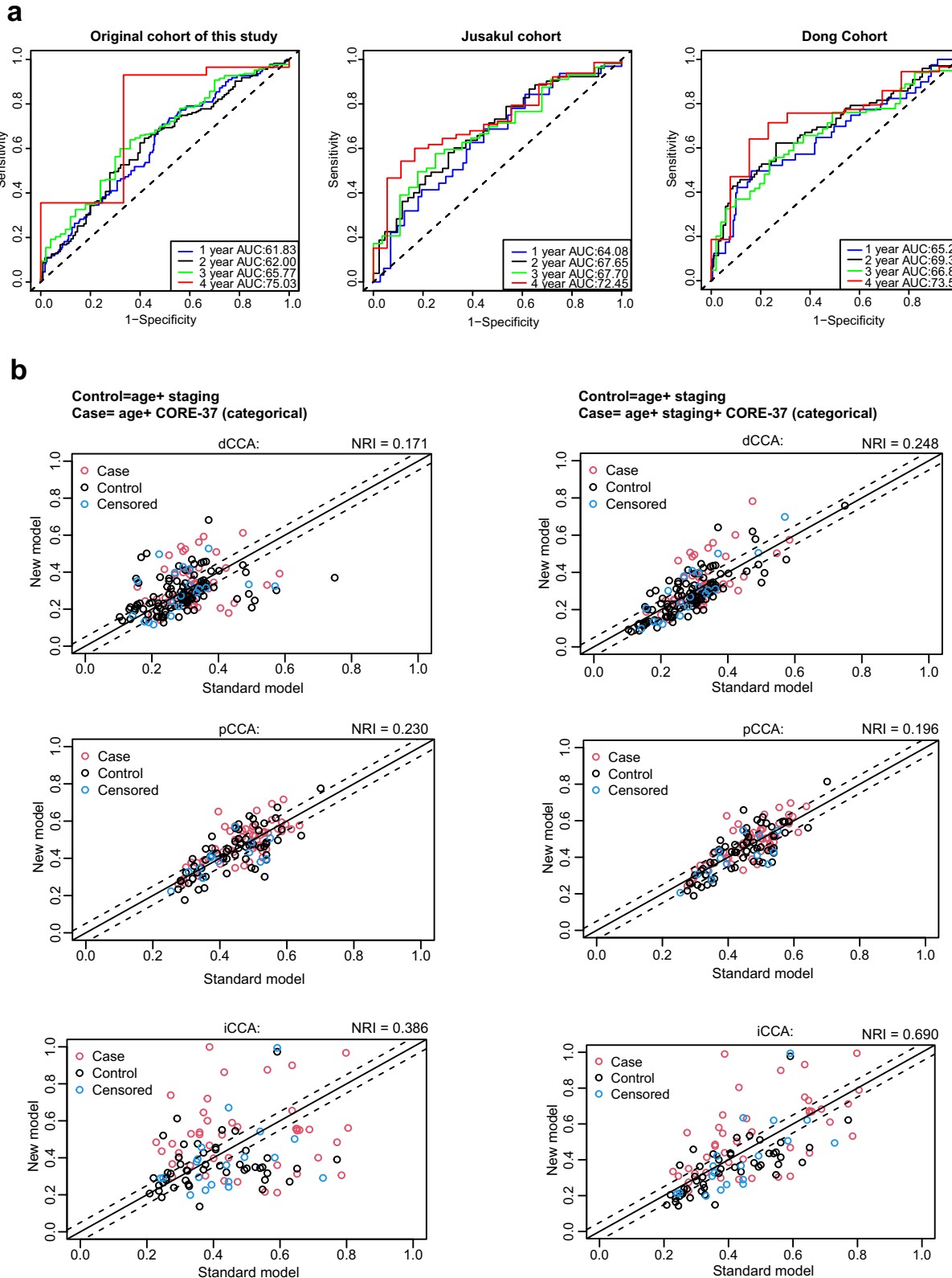

**Fig. 6 | Usefulness measurement of CORE-37 biomarker. a** ROC curves that showed time-dependent AUCs (1-4 years) based on available CORE-37 scores and clinical information of samples from our original cohort (*N* = 438), Jusakul cohort, Dong cohort, respectively. **b** Comparison of standard model (age and TNM-staging) and new models (left panel: age and CORE-37 score; right panel: age, TNM-staging and CORE-37 score) based on NRI analysis. As TNM-staging criteria vary across CCAs with different anatomical location, comparisons were performed for CCAs from the same anatomical location. ROC receiver operator characteristic. AUC area under curve, NRI net reclassification index.

predict the overall survival of CCA patients, even in samples with tissue contamination. This biomarker can be applied to assess the prognostic value of CCAs with available transcriptional profiles but vague clinical annotation.

## Discussion

Precise molecular classification of CCA tumors is a critical step, given the urgent need for targeted molecular therapy and the current limitations in anatomical-based classification[5,6,9]. While recent molecular classification schemes for CCA have been developed[6,9–11], most of them fail to consider the effect of normal tissue contamination and are only applicable to CCAs with a specific anatomical site. Thus, a classification scheme may be wrongly applied when anatomical location of a CCA tumor is ambiguous, leading to controversial results and unsatisfactory therapeutic treatment. A molecular classification scheme commonly applied to CCA independent of anatomical site may be an advantage. To solve this obstacle, we have obtained transcriptomic data of 438 clinically-annotated CCA tumors with different anatomical sites, which to our knowledge is the largest cohort investigating CCA molecular classification at transcriptomic level.

We initially identified four molecular subtypes, but found that molecular classification largely depended on tissue contamination and anatomical site. In order to reduce the impact of tissue contamination on molecular classification of CCAs and find a molecular classification scheme that independent of anatomical location, we retained a total of 164 samples as the "purified cohort", which were selected by overall tissue contamination ratio and NTP results, while the other 274 samples were used as the "verification cohort" (Fig. 1B, Table S1). Based on the analysis results of the purified cohort, we identified two universal molecular subtypes across anatomical subtypes (Fig. 4F). These two subtypes exhibit distinct pathway activities, tumor immune micro-environment and overall survival. Ferroptosis is a cell-death process driven by the oxidation of phospholipids in an iron-dependent manner[28]. Interestingly, one class also exhibits ferroptosis-resistance features and low expression levels of ferroptosis-related genes as reported in hepatocellular carcinoma[23]. In line with this, the aberrant activation of PI3K-AKT-mTOR and mTORC1 oncogenic pathways that suppress ferroptosis is also enriched[22], suggesting a critical role of this process in the progression of this CCA subtype. Thus, ferroptosis induction as a monotherapy may be less effective in treating ferroptosis-resistant C2 subtype comparing to treating C1 subtype. As ferroptosis remains largely unknown in CCA research, our findings may imply research directions regarding to this process in CCA. Overall, our results provide a comprehensive understanding of the molecular characteristics of two common CCA subtypes across different anatomical sites.

The current TNM-staging system for CCA is based on anatomical location. However, controversies exist when classifying tumors located at anatomical border areas. For example, when a tumor is located near to both the liver parenchyma and the hepatic hilum, it could be difficult to determine its origin as those at the border between the liver and hepatic hilum could only be inferred by tumor size. Additionally, whether a tumor at the junction of the common bile duct and cystic duct belongs to pCCA or dCCA is still controversial. The classification method we proposed is independent of anatomical location, thus overcomes the aforementioned limitations. Additionally, this method also has clinical application value in guiding personalized and accurate treatments.

We developed a 30-gene classifier in our purified cohort and validated our molecular classification scheme in three additional cohorts. Our results suggest that the classifier can reliably predict the molecular subtype of a CCA tumor, independent of normal tissue contamination. The predicted molecular subtypes consistently exhibited most characteristics as identified in our purified cohort, further proving the reproducibility and utility of our classification scheme. One limitation of the validation analysis is the lack of available transcriptomic data for dCCA and pCCA tumors in two external cohorts[25,26]. As our classification scheme was originally developed in cohort comprising of high number of dCCA and pCCA tumors, it still has the potential to accurately classify these tumors.

Another main contribution of our study is a prognostic biomarker named CORE-37 score that was derived from the DEGs between the two subtypes. This biomarker inferred from the expression level of 37 DEGs can stratify patients into groups with distinct survival outcomes, and is proven in both our cohort and external cohorts to be an indicator independent of sex, stage, and anatomical site. The application of CORE-37 scheme may be of great interest to clinicians for making therapeutic decision for patients with CCA.

Several limitations exist in this study. Firstly, though several potential actionable targets are recommended for each subtype based on its molecular characteristics and tumor immune microenvironment (Fig. 4f), further studies are needed to confirm these findings. Meanwhile, given that targeted treatments for CCA, such as durvalumab, pemigatinib, futibatinib, dabrafenib plus trametinib, trastuzumab plus pertuzumab, TdX and others, have already been approved in several countries, it is imperative to initiate a series of prospective studies to explore the predictive value of the CORE-37 signal in actionable targeted therapies. Secondly, it is important to note that the proportion of the three anatomical subtypes of CCA in our cohort is not evenly balanced. Minimizing the potential quantity bias associated with different anatomical subtypes during model construction could enhance the model's quality, even though the CORE-37 classification scheme is independent of anatomical subtypes. Thirdly, the number of patients in stage 4 is relatively small in our cohort, there for the predictive performance of CORE-37 in stage 4 patients still needs to be validated in larger cohorts. Last but not least, several other important characteristics may be hindered as only RNA-sequencing data is available in this study. Integrative analysis on multi-omics data may provide more comprehensive information for these subtypes (e.g., at the genomic or epigenomic level).

The conventional anatomical-based classification of cholangiocarcinoma is limited to the site contamination, that may interfere the clinical treatment decisions. Our study provides a CCA classification scheme independent of anatomical location, which can supplement the conventional anatomical-based classification and aid in treatment decision making. Our study also generates a powerful prognostic biomarker with high potential to be applied in the clinical field for predicting the survival outcome of patients with CCA.

## Methods

This study was conducted in accordance with the Declaration of Helsinki and the Ethical Guidelines for Clinical Studies, and approved by the Institutional Review Boards of The First Hospital of Jilin University (No: 2021-768).

### Patients and tumor samples

A total of 438 patients with bile duct cancer including iCCA, pCCA and dCCA, who underwent curative-intent operation between 2016 and 2021 at The First Hospital of Jilin University were involved in this study. Formalin-fixed, whole slide section paraffin-embedded (FFPE) samples were obtained from each patient. Pathological diagnosis and the estimation of non-cancer tissue area /total tissue area ratio (including hepatic, pancreatic, duodenal, lymphatic and neural tissues contamination) for each sample were done by two independent pathologists. This study was conducted in accordance with the Declaration of Helsinki and was approved by the Institutional Review Boards of The First Hospital of Jilin University (No: 21Q023-001). Informed consent was obtained from all subjects involved in the study. The clinical information of patients in this cohort is provided in Supplementary Data 1.

## RNA extraction and gene expression profiling

Total RNA was successfully extracted from 438 FFPE samples using TRIzol and RNeasy MinElute Cleanup Kit (Invitrogen). RNA purity was assessed using the NanoDrop Spectrophotometer (Thermo Fisher Scientific™, Waltham, USA). RNA integrity and concentration were measured with the RNA Nano 6000 Assay Kit of the Bioanalyzer 2100 system (Agilent Technologies, Palo Alto, CA, USA). Subsequently, mRNA libraries were created by using the NEB Next Ultra RNA Library Prep Kit (NEB, Beverly, MA, USA), following the manufacturer's protocol. Geneplus-2000 sequencing platform (Geneplus, Beijing, China) was utilized to sequence the constructed RNA-seq libraries. The sequencing reads containing adapter sequences and low-quality reads were removed to obtain high-quality reads. Reads passing quality control were aligned to the human genome hs37d5 using STAR software[29]. Transcript assembly was conducted by using StringTie2[30,31].

## Cohort stratification and molecular classification

It may not be effective and accurate enough to identify highly contaminated samples only based on normal tissue proportions inferred by anatomical judgment, as contamination degree of one sample may be subjectively under-or over-estimated. Given the most severe contaminations were hepatic or pancreas tissue, nearest template prediction (NTP) analysis was performed with R package CMScaller version 2.0.1 to determine samples showed liver-specific or pancreas-specific expression pattern[32,33], which integrated with anatomical estimation to exclude highly contaminated samples. Samples with FDR < = 0.1 in NTP result and with an overall tissue contamination proportion >25% were selected into the "verification cohort", while the other samples selected into the "purified cohort". To minimize the impact of tissue contamination on molecular classification and the following characterization, the purified cohort was used for exploration, while the verification cohort involved in validation analyses.

Non-negative matrix factorization (R package NMF version 0.24.0)[34] and consensus clustering method (R package ConsensusClusterPlus version 1.58.0)[35] were applied to obtain unsupervised molecular classes from the original and purified cohorts, respectively. To construct expression-based dendrogram, the Euclidean distance between samples was calculated and hierarchical cluster analysis was performed with the "average" cluster method.

## Gene set variation analysis and tumor microenvironment

Gene set variation analysis was performed using R package GSVA version 1.42.0[36], with method = 'ssgsea' for 50 hallmark gene sets from MSigDB collections[37] and hypoxia-related[38], and method = 'zscore' for ferroptosis-related[23] gene sets.

To investigate the tumor microenvironment, stromal and immune scores were calculated using R package ESTIMATE version 1.0.13[39]. The CIBERSORTx web tool (https://cibersortx.stanford.edu/) was applied to estimate the abundances of interested immune cells of interest in the tumor milieu[40]. Additionally, the Tumor Immune Dysfunction and Exclusion (TIDE) web tool (http://tide.dfci.harvard.edu/) was utilized to estimate the exclusion of infiltrating CD8$^+$ cytotoxic T cells[41].

## Molecular classifier design

To construct a molecular classifier, the Class Neighbors tool from the GenePattern web (https://cloud.genepattern.org/) was applied to identify genes that were closely correlated with molecular class templates[42]. Based on the impact of genes on signal-to-noise ratio (SNR) scores, genes defining each molecular class were selected to construct the classifier, which was further tested on validation cohorts with the application of NTP module from the GenePattern web (https://cloud.genepattern.org/)[42].

## Histology and immunohistochemistry (IHC)

Thirty-six representative samples from C1 ($N = 18$) and C2 ($N = 18$) were selected for tissue microarray (TMA) construction, the TMA is prepared as previously described[43]. For immunohistochemistry staining, the sections were stained using anti-Vimentin antibody (Proteintech, 1:2000), anti-Ki67 antibody (Proteintech, 1:5000), anti-Cyclin D antibody (Proteintech, 1:1500), anti- E-cadherin antibody (Proteintech, 1:2000), anti- N-cadherin antibody (Proteintech, 1:200), anti-ADH1A antibody (Abcam, 1:500), anti-CYP3A4 antibody (Proteintech, 1:200), anti-CD34 antibody (Proteintech, 1:1000), anti-VEGFA antibody (Proteintech, 1:200), anti-CD8-alpha antibody (Abcam, 1:500), anti-CD163 antibody (Proteintech, 1:1000). Images were captured by MoticEasyScan (Motic).

## Prognostic biomarker construction and validation

To construct subtype-derived prognostic biomarker, R package DESeq2 version 1.34.0 was used for gene differential expression analysis between two molecular classes[44]. Differentially expressed genes (DEGs) were extracted by filtering for adjusted $p < 0.1$ and absolute fold change>1. Top upregulated genes in molecular classes were used to construct C1-like and C2-like signatures. R package singscore version 1.14.0[45,46] was applied to compute sample-wise enrichment scores for two signatures, which outputted a unified score for the complete signature (CORE-37 score) as well as scores for C1-like and C2-like signatures separately. The sign of scores relative to the Metabolic & Proliferative-C2 signature was changed for graphical reasons.

The usefulness of CORE-37 biomarker was validated by receiver operator characteristic (ROC) curve comparison (R package timeROC version 0.4) and net reclassification index (NRI) estimation (R package nricens version 1.6). Patients with unavailable clinical information or died due to causes unrelated to cancer were excluded.

## Survival analysis

Overall survival (OS) was defined as the duration (days) between surgical resection and death of any cause or the last follow-up. Patients who died due to causes unrelated to cancer were excluded from survival analyses to enhance reliability. Survival data were analyzed by using Kaplan–Meier estimates, log-rank test, univariate and multivariate Cox regressions.

## Statistical analysis

The computational analysis, statistical analysis and plot generation in this study were performed using R software version 4.1.2 under the RStudio environment (https://www.r-project.org/). Association between categorical variables were analyzed by Fisher's exact test. Comparisons of categorical and continuous variables were done by using Kruskal–Wallis rank sum test and Wilcoxon rank sum test. All the statistical methods were used as appropriate during analyses. All reported $p$ values are two-sided and $p < 0.05$ was considered significant.

## External validation

Our constructed classifier and molecular classification scheme were tested using two external datasets: the transcript per million (TPM) matrix and clinical information from 36 patients in the TCGA-CHOL project (dCCA = 2, pCCA = 4, iCCA = 30) were downloaded at https://www.cbioportal.org;[25] the TPM matrix and clinical information from 255 patients in the Dong cohort, as well as the protein expression matrix available for 214 patients, were downloaded[26].

The Dong dataset was also used for prognostic biomarker validation. Additionally, the clinical information and expression data available for 115 patients in the Jusakul cohort (iCCA = 81, pCCA = 28, dCCA = 6) were downloaded for validating biomarker[27].

## Reporting summary

Further information on research design is available in the Nature Portfolio Reporting Summary linked to this article.

## Data availability

The raw TPM expression data of 438 samples employed in this study is provided in Supplementary Data 1. The raw sequence data reported in this paper have been deposited in the Genome Sequence Archive (GSA-Human: HRA004952) that are available under controlled access [https://ngdc.cncb.ac.cn/gsa-human/browse/HRA004952][47,48]. Access can be requested from Dr Guoyue Lv (lvgy@jlu.edu.cn). There are no specific restrictions on who may apply to for data sharing and how long the data may be used after it has been acquired.

The individual de-identified participant data (for the in-house cohort) can be made available upon request to Dr Guoyue Lv, (lvgy@jlu.edu.cn). The remaining data supporting the findings of this study are available within the article, Supplementary Information and Supplementary Data 1.

## Code availability

The source code for bioinformatics analyses can be accessed via: https://github.com/WangHaoyan6/CCA_project/blob/master/Rcode_for_github.R.

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

## Acknowledgements

We would like to thank all patients and their families for their support and for donating research samples. We would like to express our sincere gratitude to the Department of Pathology, First Hospital of Jilin University for the valuable assistance and support during the research. We thank the Department of Biobank, Division of Clinical Research, First Hospital of Jilin University for constructing tissue microarray. This study was supported by Jilin Provincial Natural Science Foundation [grant number YDZJ202201ZYTS014], National Natural Science Foundation of China [grant number 81602059], and Jilin Provincial Finance Department [grant number JLSWSRCZX2021-026].

## Author contributions

Conceptualization: G.W., Z.F.; Methodology: X.Z., J.B., Z.F., L.Z.; Software: P.W.Z., L.C., X.H., D.W., L.Z.; Investigation & Formal analysis: X.Z., J.B.; Resources & Data curation: G.W., Y.L., P.Z., F.W., X.D., M.W., X.S., B.J.; Writing - Original Draft & Validation: X.Z.; Writing – Review & Editing: Z.F., X.X., J.B., H.W.; Supervision: G.L., J.B., X.F.X., G.W., Y.L.; Project administration: Y.J., X.F.X., X.X., X.Y.; Funding acquisition: Z.F.

## Competing interests

The authors declare no competing interests.
