## [Peer Review File · Nature Communications]

REVIEWER COMMENTS

Reviewer #1 (Remarks to the Author): expertise in CCA subtypes and tumour microenvironment

Fan et al. performed RNA-seq on a large cohort of cholangiocarcinoma (CCA) and proposed a novel CCA classification, in which two subtypes displayed distinct molecular characteristics and survival outcomes. Also, the authors developed a 30-gene classifier and a novel biomarker to predict the molecular subtype of CCA and prognosis, respectively, and the results were validated in external cohorts. However, the study is almost descriptive and the evidence for some conclusions is not enough, which may require further experimental validation. Some important issues that need to be addressed are in the comments below.

Major comments:

1. Figure 2, the results of angiogenesis and immune infiltration need to be validated by immunohistochemistry.
2. Figure 4F, the data from cell and murine experiments can help the conclusion be more reliable.
3. Figure 5B, 23.2% (38/164) of patients can not be divided into a certain molecular subtype using nearest template prediction (NTP) with 30-gene classifier. Perhaps other methods can fix this.
4. Figure 5C, unclear the significance of CORE-37 score since the authors have developed a 30-gene classifier.
5. The study needs more in-depth analyses. E.g., the comparison with previous molecular subtypes and others.

Minor comments:

1. Arial font should be used in the figures.
2. The author is encouraged to refine and enhance the quality of the figures presented in the manuscript.
3. Figure 1, why use three methods of clustering (non-negative matrix factorization, hierarchical clustering, and consensus clustering)?
4. Figure 2A, why the scores of different hallmarks are the same?
5. There are some errors in the manuscript. E.g., "characteristics" should be added behind "molecular" (line 55). "C3" and "C4" should be reversed (line 130). "C1" should be "C4" (line 132).

Reviewer #2 (Remarks to the Author): expertise in cholangiocarcinoma transcriptomics

Fan et al. reported a novel molecular classification of cholangiocarcinoma (CCA) based on gene expression analysis of FFPE samples and investigated its prognostic significance using several CCA cohorts. They first selected a purified cohort of CCA cases with less normal tissue contamination and identified two molecular subclasses that showed distinct biological pathways, immunological environment and prognosis. These subclasses were reproducible in additional cohorts of cases with more normal tissue contamination. They then extracted a 30-gene classifier (CORE-37) of the two subtypes, which is a prognostic biomarker independent of anatomical location and tumor stage.

General comments

It is interesting but a little skeptical to see that CCA can be classified into the two subtypes independent of anatomical location because genetic data showed that driver mutations are diverse among tumors in distinct anatomical locations. Therefore, the association of genetic data with these molecular subtypes is essential.

1. Classification of all 438 cases

The authors claimed that the initial four subtypes (C1-4) were significantly associated with anatomical location and normal contamination. Fig. S1 shows that three groups would also be good classifiers. How this three-subtype classification was associated with anatomical location and normal contamination?

2. Estimation of normal contamination

The authors should show the pathological verification of their normal tissue-contaminated cases. Generally normal pancreatic tissue contains mostly islet cells and it was uncertain why such pancreatic tissue is contaminated in CCA.

Fig 1A/D also showed lymphatic/neural contamination data. How the authors precisely evaluated these contaminants?

The Dendrogram of Fig 1A/D should also be presented to see the classification of cases.

The detailed results of the NTP analysis should be shown in the Supplementary data.

Because the ICGC cohort contains mutation data, the authors should verify whether their estimation is comparable to the genetic evaluation of non-tumor contamination.

3. Immunological environments

If the authors claimed that normal tissue contamination is problematic for extracting "pure" molecular subtypes, why did they ignore immune cell contamination? The authors should also exclude cases with high immune cell contaminated cases for classification.

They may examine the relationship between molecular subtypes and immune environments using a verified (contaminated) cohort because they would contain more immune cells.

4. Association with genetic data

ICGC dataset has mutation data, and the authors should examine how the diversity of driver mutations can be classified into simple two subtypes.

Minor comments

1. Figure 1C is very difficult to see. It should be much improved.
2. The NRI value is better presented in a non-supplementary table because it is difficult to assess the superiority of Figure 6B.

Reviewer #3 (Remarks to the Author): clinical expertise in liver cancer

We congratulate the authors for the work and findings.
Some additional points raised on the revision should be accessed:

Results:

- 1) Line 310: 'However, divergencies partially occurred in TCGA-CHOL cohort and Dong cohort possibly due to bias effect of dominated iCCAs in both cohorts'.

Authors should better explain this divergences, considering that the signature evaluated is anatomical independent, the influence of iCCA would not be important. Also, this is a clear limitation of the signature evaluated, and this limitation should be included in the discussion. iCCA is only 16% of the purified cohort.

- 2) Line 374: 'In summary, the CORE-37 score derived from the molecular subtype signatures is a highly reliable prognostic indicator of CCAs, regardless of stage and anatomical location of tumors. These results confirm the predictive performance and universality of CORE-37 score..'

Authors should re-evaluate this sentence. We don't have any data about treatments performed included radiotherapy, chemotherapy or targeted treatments. It is not possible to say that CORE-37 is a predictive platform. It is only prognostic. If the study aims to classify the platform as predictive will need to evaluate the impact of CORE-37 and systemic treatments performed in the patients, including platinum therapy or immunotherapy.

3) We can see that stage 4 disease is rare in the cohorts dCCA and pCCA, this can cause a limitation of the findings that the CORE-37 is superior to TNM staging, due to this imbalance. The sentence: 'The results showed positive NRI values in all three comparisons, demonstrating that the prediction performance of CORE-37 prognostic biomarker was superior to that of TNM staging, regardless of anatomical location'.., could not be assumed considering that stage 4 is just 2 cases in dCCA and pCCA cohorts and 6 in iCCA cohort. This signature is validated mostly in resected patients (>80=90% of the overall cohort), this is a limitation of the findings and should be discussed.

Discussion:

1) " Our study also generates a powerful prognostic biomarker with high potential to be applied in the clinical field for predicting the survival outcome of patients with CCA" ..

It is important to include in the discussion/conclusion that targeted treatments are already approved for CCA in several countries including durvalumab, pemigatinib, futibatinib, dabrafenib +trametinib, trastuzumab +pertuzumab, TdX and others. If CORE-37 signature is more valuable in clinical practice in the setting of advanced stages than actionable targets is yet unknown, and prospective studies evaluating systemic treatments performed are necessary to confirm this findings..

RESPONSE TO REVIEWERS' COMMENTS

Reviewer #1 (Remarks to the Author): expertise in CCA subtypes and tumour microenvironment

Fan et al. performed RNA-seq on a large cohort of cholangiocarcinoma (CCA) and proposed a novel CCA classification, in which two subtypes displayed distinct molecular characteristics and survival outcomes. Also, the authors developed a 30-gene classifier and a novel biomarker to predict the molecular subtype of CCA and prognosis, respectively, and the results were validated in external cohorts. However, the study is almost descriptive and the evidence for some conclusions is not enough, which may require further experimental validation. Some important issues that need to be addressed are in the comments below.

Major comments:

1. Figure 2, the results of angiogenesis and immune infiltration need to be validated by immunohistochemistry.

Thanks for your comment. In order to more effectively mitigate batch effects caused by staining, we carefully chose thirty-six representative samples from C1 (N=18) and C2 (N=18) for the construction of tissue microarray (TMA) and immunohistochemistry (IHC) staining.

We have included the following explanation in the manuscript:

“CD163 was used as a marker for M2 macrophages, and it was shown that CD163 expression in C1 was significantly higher than that in C2. On the other hand, CD8 T cells were found to be more abundant in C2; However, this difference did not reach statistical significance ($p=0.08$), possibly due to small sample size.” (Line 218-222)

Furthermore, we used CD34 as a marker for angiogenesis and observed a trend toward stronger angiogenesis in C1 CCA compared to C2 CCA, although this difference was not statistically significant ($p=0.12$). Additionally, we noted that VEGFA was predominantly expressed in tumor cells; unfortunately, we did not detect any significant differences in VEGFA protein levels between the C1 and C2 groups. This could be attributed to the small variations in VEGFA protein levels and the semi-quantitative nature of IHC quantification per se. Overall, our TMA IHC results are consistent with the RNA level findings, providing substantial evidence for the robustness of our CCA classification. The results have been included in supplementary Figure 5.

2. Figure 4F, the data from cell and murine experiments can help the conclusion be more reliable.

Thanks for your suggestions. We genuinely value your feedback, and recognize the crucial role of cellular and animal experiments in drawing definitive conclusions in biological research. However, conducting research in cholangiocarcinoma presents specific challenges. The availability of commercially obtainable cell lines for cholangiocarcinoma is extremely limited, and there is currently no well-established spontaneous cholangiocarcinoma mouse model.

While it would be ideal to find cell lines that faithfully represent the C1 and C2 subtypes for subsequent cellular experiments, categorizing them solely based on their sequencing data is

not feasible. A more practical approach may involve isolating cells from fresh tissues classified as C1 or C2 and conducting organoid or primary cell culture. We are committed to considering the incorporation of this aspect into our future work.

We kindly request Reviewer#1 to permit us to forego additional cellular and animal experiments in this study. To compensate for this limitation, we employed IHC on TMA to validate the expression of key proteins used in our classification, thus affirming the validity of our classification at the protein level. These proteins include proliferation-associated proteins Ki67 and Cyclin D, both of which exhibited lower expression levels in C1 compared to C2. We have included the following explanation in the manuscript:

“E-cadherin was significantly lower in C1 than in C2, while Vimentin displayed higher expression in C1 tumor cells than in C2, suggesting a more pronounced epithelial-mesenchymal transition in C1 compared with C2, although N-cadherin did not show significant differences between the two groups.” (Line 192-196)

We also examined the metabolic changes using ADH1A and CYP3A4 as markers. We have included the following explanation in the manuscript:

“ADH1A and CYP3A4 were used as markers to detect metabolic changes by immunohistochemistry (IHC). Consistent with the transcriptome analysis results, we found that both ADH1A and CYP3A4 were significantly lower in C1 compared to C2 (Fig. S5).” (Line 238-241)

The results have been incorporated into supplementary Figure 4 and 5.

Once again, we sincerely appreciate your valuable advice

3. Figure 5B, 23.2% (38/164) of patients can not be divided into a certain molecular subtype using nearest template prediction (NTP) with 30-gene classifier. Perhaps other methods can fix this.

The following figure present the comparison of our molecular classification results and those obtained from other methods, using our pure cohort samples. This analysis includes the work of Montal et al. (*Montal R, Sia D, Montironi C, et al. Molecular classification and therapeutic targets in extrahepatic cholangiocarcinoma. J Hepatol. 2020;73(2):315-327. doi: 10.1016/j.jhep.2020.03.008. PMID: 32173382; PMCID: PMC84189;*) and Sia et al. (*Sia D, Hoshida Y, Villanueva A, et al. Integrative molecular analysis of intrahepatic cholangiocarcinoma reveals 2 classes that have different outcomes. Gastroenterology. 2013;144(4):829-840. doi:10.1053/j.gastro.2013.01.001.*)

The results showed that 12.8% (21/164) samples could not be classified into specific subtypes using Montal et al.'s molecular classification method, which was developed for extrahepatic cholangiocarcinoma (eCCA); additionally, 71.3% (117/164) samples could not be classified into specific subtypes using Sia et al.'s molecular classification method, which was designed for intrahepatic cholangiocarcinoma (iCCA). Regardless of the methods employed, there remain some samples that defy classification, particularly when utilizing Sia et al.'s molecular classification method. This observation underscores the need for establishing an anatomy-independent molecular classification method.

4. Figure 5C, unclear the significance of CORE-37 score since the authors have developed a 30-gene classifier.

Thank you for pointing out the areas in the manuscript that require further clarification. We have included the following explanation in the manuscript:

“As the 30-gene classifier divides samples into two categories, quantifying the relationship between this categorical variable and survival outcomes can be challenging. To address this issue, we sought to identify a continuous variable as a prognostic biomarker derived from subtypes, the value of which is related to survival outcomes in CCA patients.” (Line 342-346)

Consequently, we developed the CORE-37 score based on the differentially expressed genes (DEGs) from C1 and C2, subtypes and explored the potential utility of the CORE-37 score.

5. The study needs more in-depth analyses. E.g., the comparison with previous molecular subtypes and others.

The comparison of our molecular classification results and those obtained from other methods (using our pure cohort samples) is presented in response to major comment 3. In brief, when using the methods of Montal et al. or Sia et al. for classification, we encountered challenges with classifying 12.8% (21/164) or 71.3% (117/164) of the samples, respectively. This discrepancy arises because the molecular classification methods employed by others

were developed based on a single CCA anatomical subtype. In contrast, our cohort comprises not just one but a combination of anatomical subtype, including 32.32% dCCA, 51.22% pCCA, and 16.46% iCCA. Therefore, when applying molecular classification models constructed by others that are tailored to a specific CCA anatomical subtype, we encounter difficulties in obtaining meaningful and interpretable results.

Minor comments:

1. Arial font should be used in the figures.

We have made the requested modifications to ensure that Arial font is now consistently used throughout the figures, as per your suggestion.

2. The author is encouraged to refine and enhance the quality of the figures presented in the manuscript.

We appreciate your feedback and have fully committed to improving the clarity, resolution, and overall presentation of the figures to ensure they meet the highest standards of quality.

3. Figure 1, why use three methods of clustering (non-negative matrix factorization, hierarchical clustering, and consensus clustering)?

We aim to demonstrate through various means that our classification is not associated with anatomical subtypes

4. Figure 2A, why the scores of different hallmarks are the same?

Single-sample gene set enrichment analysis (ssGSEA) was used to obtain enrichment scores, with samples from the same subtype indicated with a normalized z-score.

5. There are some errors in the manuscript. E.g., "characteristics" should be added behind "molecular" (line 55). "C3" and "C4" should be reversed (line 130). "C1" should be "C4" (line 132).

Thank you for pointing out the errors in our manuscript. We have addressed these issues and made the necessary corrections accordingly.

Reviewer #2 (Remarks to the Author): expertise in cholangiocarcinoma transcriptomics

Fan et al. reported a novel molecular classification of cholangiocarcinoma (CCA) based on gene expression analysis of FFPE samples and investigated its prognostic significance using several CCA cohorts. They first selected a purified cohort of CCA cases with less normal tissue contamination and identified two molecular subclasses that showed distinct biological pathways, immunological environment and prognosis. These subclasses were reproducible in additional cohorts of cases with more normal tissue contamination. They then extracted a 30-gene classifier (CORE-37) of the two subtypes, which is a prognostic biomarker independent of anatomical location and tumor stage.

General comments

It is interesting but a little skeptical to see that CCA can be classified into the two subtypes independent of anatomical location because genetic data showed that driver mutations are diverse among tumors in distinct anatomical locations. Therefore, the association of genetic data with these molecular subtypes is essential.

1. Classification of all 438 cases

The authors claimed that the initial four subtypes (C1-4) were significantly associated with anatomical location and normal contamination. Fig. S1 shows that three groups would also be good classifiers. How this three-subtype classification was associated with anatomical location and normal contamination?

The most common approach for determining the best k value involves utilizing the cophenetic correlation coefficient. As proposed by Brunet et al. (Proc Natl Acad Sci U S A. 2004;101(12):4164-4169.), it is recommended to select the smallest value of 'r' at which this coefficient begins to decline. According to our results, the cophenetic coefficient remained notably high for both k=2 and k=4, whereas it commenced a greater number of classes, we determined that k=4 represented the most suitable option. This approach aligns with Burnet et al.'s recommendation while also addressing our specific objective of achieving a more refined classification.

2. Estimation of normal contamination

The authors should show the pathological verification of their normal tissue-contaminated cases. Generally normal pancreatic tissue contains mostly islet cells and it was uncertain why such pancreatic tissue is contaminated in CCA.

Thanks for your valuable comments and suggestions. You have raised a very important point. Indeed, normal pancreatic tissue mainly comprises islet cells and glandular cells. However, it is critical to note that the extrahepatic bile duct traverses the pancreas, entering the duodenal papilla to facilitate bile excretion into the intestines. Additionally, the main pancreatic duct also joins the duodenum. It is not uncommon for extrahepatic CCA, especially distal CCA, to invade the pancreas. To distinguish whether the tumor originates from the pancreas or the bile duct, the pathology department mandate the preservation of a portion of the pancreas during sampling.

Due to the presence of this situation, contamination of pancreatic tissue can occur during

the application of RNA sequencing on paraffin samples. To illustrate this issue, we have compiled typical contamination images, which are provided in supplementary Figure 1.

Fig 1A/D also showed lymphatic/neural contamination data. How the authors precisely evaluated these contaminants?

Thanks for your comment. In the field of pathological diagnosis, especially in immunohistochemistry, pathologists typically determine tissue proportions based on visual inspection rather than relying on automated evaluation. Additionally, in our research, the assessment of tissue proportions is conducted by experienced senior pathologists. Therefore, we believe that these results adequately reflect the differences in normal tissue contamination among the various samples. As evident in supplementary Figure 1 D/E, neural and lymphatic contaminations are respectively observed in the pathological sections.

The Dendrogram of Fig 1A/D should also be presented to see the classification of cases.

Given that Figures 1A and D are presented in the order of molecular classification, it is not feasible to incorporate dendrogram to illustrate classification. Figure 1A shows the unsupervised cluster analysis of transcriptomic data using the NMF method, highlighting the significant impact of normal tissue contamination on molecular classification results. For instance, hepatic tissue predominates in C4, while pancreatic tissue is primarily observed in C2.

Subsequently, we introduced the “purified cohort”, characterized by reduced normal tissue contamination. Figure 1C demonstrates hierarchical clustering analysis on this purified cohort, revealing that CCAs with minimal contamination failed to be distinguished by anatomical location at the transcriptomic level, and samples from various anatomical sites did not cluster together.

As a solution, we attempted to classify the purified cohort through consensus clustering as depicted in Figure 1D. Notably, in the purified cohort, normal tissue contamination had a minimal impact on molecular classification. This forms the basis for the subsequent development of an anatomy-independent molecular classification scheme of CCA.

The detailed results of the NTP analysis should be shown in the Supplementary data.

Thank you for your advice. We have added the results of NTP analysis to the Supplementary Table 6.

Because the ICGC cohort contains mutation data, the authors should verify whether their estimation is comparable to the genetic evaluation of non-tumor contamination.

As the reviewer mentioned, genomic data can be utilized to estimate the normal tissue contamination. Various bioinformatics tools and software packages are available for estimating tumor purity and normal tissue contamination. Tools like ABSOLUTE, FACETS, and Sequenza can be helpful in this regard. These tools often incorporate both mutation and copy number variations (CNV) information. Following the reviewer's suggestion, we conducted a search in the ICGC database and retrieved two projects that exclusively provided mutation data.

ICGC Data Portal

P

BILIARY TRACT CANCER - JP

Summary

Donors

Mutations

Summary

Code	BTCA-JP
Name	Biliary Tract Cancer - JP
Primary Site	Gall Bladder
Tumour Type	Biliary Tract cancer
Tumour Subtype	Multiple histological subtypes
Countries	Japan
Number of donors with molecular data in DCC	239
Total number of donors	239

Available Data Types:

Clinical Data	239 donors
Simple Somatic Mutations (SSM)	239 donors
Copy Number Somatic Mutations (CNSM)	--
Structural Somatic Mutations (SSM)	--
Simple Germline Variants (SGV)	--
Array-based DNA Methylation (METH-A)	--
Sequence-based DNA Methylation (METH-S)	--
Array-based Gene Expression (EXP-A)	--
Sequence-based Gene Expression (EXP-S)	--
Protein Expression (PEXP)	--
Sequence-based miRNA Expression (miRNA)	--
Exon Junction (JCEN)	--

Open in DCC Data Releases
View in Data Repositories

ICGC Data Portal

P

BILIARY TRACT CANCER - SG

Summary

Donors

Mutations

Summary

Code	BTCA-SG
Name	Biliary Tract Cancer - SG
Primary Site	Gall Bladder
Tumour Type	Biliary Tract cancer
Tumour Subtype	Gall bladder cancer / Cholangiocarcinoma
Countries	Singapore
Number of donors in PCAWG	12
Number of donors with molecular data in DCC	71
Total number of donors	71

Available Data Types:

Clinical Data	71 donors
Simple Somatic Mutations (SSM)	71 donors
Copy Number Somatic Mutations (CNSM)	--
Structural Somatic Mutations (SSM)	--
Simple Germline Variants (SGV)	--
Array-based DNA Methylation (METH-A)	--
Sequence-based DNA Methylation (METH-S)	--
Array-based Gene Expression (EXP-A)	--
Sequence-based Gene Expression (EXP-S)	--
Protein Expression (PEXP)	--
Sequence-based miRNA Expression (miRNA)	--
Exon Junction (JCEN)	--

Open in DCC Data Releases
View in Data Repositories

Regrettably, the absence of CNV data in these projects hinders our ability to accurately infer tumor cell purity. Additionally, both of these projects lack the expression data essential for conducting Nearest Template Prediction (NTP) analysis. Therefore, even if genomic data could be employed to estimate normal tissue contamination, we would be unable to carry out orthogonal validation.

3. Immunological environments

If the authors claimed that normal tissue contamination is problematic for extracting "pure" molecular subtypes, why did they ignore immune cell contamination? The authors should also exclude cases with high immune cell contaminated cases for classification. They may examine the relationship between molecular subtypes and immune environments using a verified (contaminated) cohort because they would contain more immune cells.

On the one hand, we consider the infiltration of immune cells to be a vital component of the tumor microenvironment, and we believe that samples containing a significant degree of immune cell infiltration should not be haphazardly removed. Tertiary lymphoid structures, which are rich in immune cells, are integral to the tumor microenvironment and should be

retained.

On the other hand, our classification results reveal that the Mesenchymal & Immunosuppressive-C1 subtype exhibits immunosuppressive characteristics compared to Metabolic & Proliferative-C2. Excluding samples with high immune infiltration may lead to loss of distinguishing features between these two subtypes.

Furthermore, when establishing the “purified cohort”, we excluded samples with lymph node contamination proportions >25%. Consequently, the purified cohort does not include samples with substantial lymph node contamination.

4. Association with genetic data

ICGC dataset has mutation data, and the authors should examine how the diversity of driver mutations can be classified into simple two subtypes.

Regrettably, due to the absence of genome-level sequencing for our samples, we are unable to construct a classification model based on genomic data. In addition, the mutation data of ICGC cannot be integrated to the molecular classification model established based on transcriptome data. Consequently, we are presently unable to classify and explore the mutation data from the ICGC database.

This situation underscores the importance of conducting multi omics research based on our molecular classification. We have also emphasized the significance of analyzing genomics and epigenetics data in the discussion.

Minor comments

1. Figure 1C is very difficult to see. It should be much improved.

We have bolded the font in Figure 1C and improved the quality of all figures. Thank you for your advice.

2. The NRI value is better presented in a non-supplementary table because it is difficult to assess the superiority of Figure 6B.

Thank you for your suggestion. We have included the NRI value in Figure 6B.

Reviewer #3 (Remarks to the Author): clinical expertise in liver cancer

We congratulate the authors for the work and findings.

Some additional points raised on the revision should be accessed:

Results:

1) Line 310: 'However, divergencies partially occurred in TCGA-CHOL cohort and Dong cohort possibly due to bias effect of dominated iCCAs in both cohorts'.

Authors should better explain this divergences, considering that the signature evaluated is anatomical independent, the influence of iCCA would not be important. Also, this is a clear limitation of the signature evaluated, and this limitation should be included in the discussion. iCCA is only 16% of the purified cohort.

Thank you for identifying the inaccuracies in the manuscript. We have implemented the following revisions in the relevant sections of the manuscript:

“However, divergencies were observed in TCGA-CHOL cohort and Dong cohort (Fig. S10, Fig. S12). These discrepancies may be attributed, in part, to the bias introduced by tissue contamination in both cohorts.” (Line 322-325)

We also included the discussion about the abundant proportion of dCCA and pCCA in the cohort:

“One limitation of the validation analysis is the lack of available transcriptomic data for dCCA and pCCA tumors in two external cohorts [31,32]. As our classification scheme was originally developed in cohort comprising of high number of dCCA and pCCA tumors, it still has the potential to accurately classify these tumors.” (Line 451-455)

2) Line 374: 'In summary, the CORE-37 score derived from the molecular subtype signatures is a highly reliable prognostic indicator of CCAs, regardless of stage and anatomical location of tumors. These results confirm the predictive performance and universality of CORE-37 score..'

Authors should re-evaluate this sentence. We don't have any data about treatments performed included radiotherapy, chemotherapy or targeted treatments. It is not possible to say that CORE-37 is a predictive platform. It is only prognostic. If the study aims to classify the platform as predictive will need to evaluate the impact of CORE-37 and systemic treatments performed in the patients, including platinum therapy or immunotherapy.

Thank you for pointing out the imprecise portions of the manuscript. We have addressed this concern by making the following modifications to the corresponding section of the manuscript:

“These results validate the prognostic performance and applicability of CORE-37 score.” (Line 391-392)

We appreciate your meticulous review and valuable feedback.

3) We can see that stage 4 disease is rare in the cohorts dCCA and pCCA, this can cause a limitation of the findings that the CORE-37 is superior to TNM staging, due to this

imbalance.

The sentence: 'The results showed positive NRI values in all three comparisons, demonstrating that the prediction performance of CORE-37 prognostic biomarker was superior to that of TNM staging, regardless of anatomical location'.., could not be assumed considering that stage 4 is just 2 cases in dCCA and pCCA cohorts and 6 in iCCA cohort.

This signature is validated mostly in resected patients (>80=90% of the overall cohort), this is a limitation of the findings and should be discussed.

Thank you for pointing out the parts of the manuscript that require further discussion. We have included the following discussion in the manuscript:

" Secondly, it is important to note that the proportion of the three anatomical subtypes of CCA in our cohort is not evenly balanced. Minimizing the potential quantity bias associated with different anatomical subtypes during model construction could enhance the model's quality, even though the CORE-37 classification scheme is independent of anatomical subtypes." (Line 471-476)

Discussion:

1) " Our study also generates a powerful prognostic biomarker with high potential to be applied in the clinical field for predicting the survival outcome of patients with CCA"..

It is important to include in the discussion/conclusion that targeted treatments are already approved for CCA in several countries including durvalumab, pemigatinib, futibatinib, dabrafenib +trametinib, trastuzumab +pertuzumab, TdX and others. If CORE-37 signature is more valuable in clinical practice in the setting of advanced stages than actionable targets is yet unknown, and prospective studies evaluating systemic treatments performed are necessary to confirm this findings..

Thank you for pointing out the parts of the manuscript required further discussion. We have included the following discussion in the manuscript:

" Meanwhile, given that targeted treatments for CCA, such as durvalumab, pemigatinib, futibatinib, dabrafenib plus trametinib, trastuzumab plus pertuzumab, TdX and others, have already been approved in several countries, it is imperative to initiate a series of prospective studies to explore the predictive value of the CORE37 signal in actionable targeted therapies." (Line 466-471)

REVIEWERS' COMMENTS

Reviewer #1 (Remarks to the Author):

The authors have addressed my concerns.

Reviewer #2 (Remarks to the Author):

The authors responded to part of my questions, but they didn't exactly answer some.

1. Classification of all 438 cases

How this three-subtype classification was associated with anatomical location and normal contamination?

2. Association with genetic data

As the authors classified ICGC samples into the two subtypes, it is not difficult to see any driver mutations (not CNV) associated with them.

Reviewer #3 (Remarks to the Author):

The authors have addressed all the clinical questions raised.

The work brings significance to the field adding new insights in the molecular classification.

RESPONSE TO REVIEWERS' COMMENTS

Reviewer #1 (Remarks to the Author):

The authors have addressed my concerns.

Reviewer #2 (Remarks to the Author):

The authors responded to part of my questions, but they didn't exactly answer some.

1. Classification of all 438 cases

How this three-subtype classification was associated with anatomical location and normal contamination?

When the 438 samples were divided into three subtypes using the NTP method, similar to the four subtypes in our manuscript, normal tissue contamination and anatomical location dominated the clustering of the samples.

This result shows significantly enriched iCCAs in C1 and dCCAs in C2. Normal tissue contamination significantly impacted molecular classification, with hepatic tissue more predominant in C1 and pancreatic tissue in C2. The subtype preference was also reflected in the distribution of duodenal, lymphatic and neural tissues. The anatomical location of the lesions determines the surgical approach and the extent of tissue contamination. This again confirms the necessity of molecular typing after excluding normal tissue contamination.

2. Association with genetic data

As the authors classified ICGC samples into the two subtypes, it is not difficult to see any driver mutations (not CNV) associated with them.

As we mentioned in the last round of responses, tools like ABSOLUTE, FACETS, and Sequenza often incorporate both mutation and copy number variations (CNV) information. Unfortunately, the absence of CNV data in two projects of ICGC database impede our ability to accurately infer tumor cell purity. In addition, both projects lack the expression data necessary to perform Nearest Template Prediction (NTP) analysis. Thus, even if genomic data could be employed to estimate normal tissue contamination, we could neither be able to classify ICGC samples into two subtypes nor explore the mutation of ICGC samples.

Reviewer #3 (Remarks to the Author):

The authors have addressed all the clinical questions raised.

The work brings significance to the field adding new insights in the molecular classification.